# Optimal Auction Design in the Joint Advertising

**Yang Li** [1]   **Yuchao Ma** [1]   **Qi Qi** [†1 2 3]

## Abstract

Online advertising is a vital revenue source for major internet platforms. Recently, joint advertising, which assigns a bundle of two advertisers in an ad slot instead of allocating a single advertiser, has emerged as an effective method for enhancing allocation efficiency and revenue. However, existing mechanisms for joint advertising fail to realize the optimality, as they tend to focus on individual advertisers and overlook bundle structures. This paper identifies an optimal mechanism for joint advertising in a single-slot setting. For multi-slot joint advertising, we propose **BundleNet**, a novel bundle-based neural network approach specifically designed for joint advertising. Our extensive experiments demonstrate that the mechanisms generated by **BundleNet** approximate the theoretical analysis results in the single-slot setting and achieve state-of-the-art performance in the multi-slot setting. This significantly increases platform revenue while ensuring approximate dominant strategy incentive compatibility and individual rationality.

## 1. Introduction

Online advertising emerges as the principal revenue source for internet platforms like Google, Amazon, and Facebook. With online advertising revenue reaching approximately $225 billion in 2023, it plays a critical role in sustaining the growth and development of these platforms. A prevalent method for allocating advertisement slots is sponsored search auctions, wherein advertisers submit bids to the platform. The platform subsequently employs predefined auction mechanisms to determine ad placements and pricing. Maximizing the monetization efficiency of advertising traf-

fic thus becomes a fundamental research direction for these companies.

In recent years, a novel auction scenario known as "joint advertisement" emerge on online advertising platforms such as Facebook (Facebook, 2024). This joint advertisement scenario (Ma et al., 2024), where both stores and brands contribute to a bundle for advertising, allows for a more integrated approach to advertisement placement. As illustrated in Figure 1, in traditional advertising pages, each product is bid on solely by the retailer as a single bidder. However, in joint advertisement, the retailer and the supplier jointly bid on an ad. The ranking of the advertisement is thus influenced by the combined bid rather than by a single party's bid, and the platform can charge both participants accordingly. By accommodating the interests of multiple stakeholders—including platforms, retailers, and brand suppliers—joint advertisements create a mutually beneficial ecosystem, fostering a win-win situation for all parties involved.

Existing approaches for improving revenue in joint advertisements can be categorized into two types: the first is the use of Vickrey-Clark-Groves (VCG) mechanism (Vickrey, 1961; Clarke, 1971; Groves, 1973; Ma et al., 2024) and VCG-like mechanisms such as JAMA (Ma et al., 2024), and the second is the application of automated mechanism design methods to joint advertisements, known as JRegNet (Zhang et al., 2024). Compared to VCG, JAMA struggles to adapt to the flexible and complex bipartite relationships between retailers and suppliers that often arise in joint advertisement scenarios. JRegNet, on the other hand, suffers from poor model generalization and robustness, and in many cases, its performance is inferior to VCG, leading to reduced rather than increased revenue.

Nevertheless, these methods have the following issues:

- These methods are similar to traditional methods in that they only establish incentive compatibility (IC) in the brand (or store) dimension. However, in the scenario of joint advertising, the final ad slot will be allocated to a bundle consisting of a brand and a store. Previous work did not fully characterize the bundle, resulting in space for optimization in the actual ad slot allocation process.

- In addition, since these methods cannot ensure the perfect

[1]Gaoling School of Artificial Intelligence, Renmin University of China, Beijing, China [2]Beijing Key Laboratory of Research on Large Models and Intelligent Governance [3]Engineering Research Center of Next-Generation Intelligent Search and Recommendation, MOE. Correspondence to: Qi Qi <qi.qi@ruc.edu.cn>.

*Proceedings of the 42^{nd} International Conference on Machine Learning*, Vancouver, Canada. PMLR 267, 2025. Copyright 2025 by the author(s).

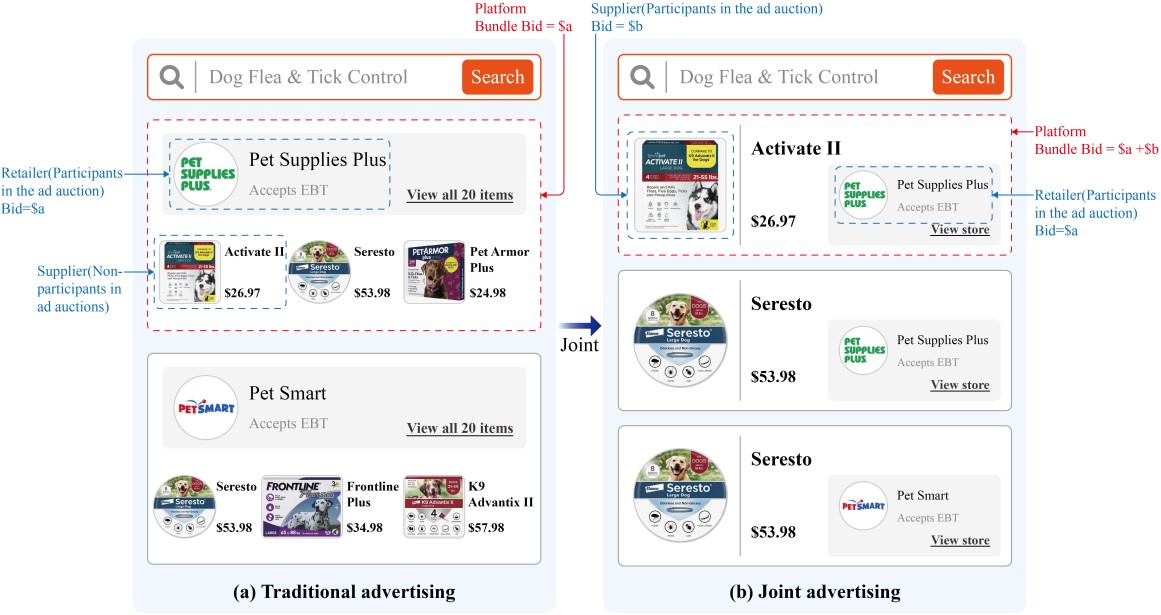

*Figure 1.* A Comparison Between the Traditional Advertising Model and the Joint Advertising Model. The image on the left illustrates traditional advertising, where the retailer submits a bid for the ad, while the supplier (brand owner) does not participate in the auction. In this case, the platform only receives the bid from the retailer. In contrast, as shown in the right image, for a joint advertisement, both the retailer and the supplier participate in the auction. They submit bids simultaneously, and the platform receives bids from both parties.

fulfillment of IC when maximizing the platform's revenue, the regret value is used to measure the degree of IC violation. Although the lower the regret value, the closer the mechanism is to the optimal, it may not be close to the optimal in the parameter space, so there is the possibility of a locally optimal solution.

To address these challenges, we propose a novel and efficient automated mechanism learning approach, termed **BundleNet**, specifically tailored for joint advertisements.

Concretely, the contributions of our paper can be summarized as follows:

First, we identify the optimal mechanism for single-slot joint advertisement. Additionally, we propose a novel neural network architecture and introduce a new IC constraint method for multi-slot joint advertisements. Our approach not only improves platform revenue but also ensures approximate IC and individual rationality (IR). Extensive experiments demonstrate that our method achieves state-of-the-art performance.

## 2. Related Work

In traditional advertising auctions, the generalized second price (GSP) auction, following early work (Aggarwal et al., 2006; Varian, 2007), has been popular in the past two decades for its simplicity, feasibility, and good revenue

(Edelman et al., 2007). Variations like adding reserved prices (Thompson & Leyton-Brown, 2013; Roberts et al., 2016) and "squashing" (Lahaie & Pennock, 2007; Charles et al., 2016) have been introduced to boost revenue. But GSP lacks the IC property.

In auction mechanism design, Myerson (Myerson, 1981) characterized the revenue maximizing single-parameter auction mechanism. However, the multi-parameter optimal auction design problem remains unsolved even after 40 years. Automated mechanism design (Conitzer & Sandholm, 2002; 2004; Sandholm & Likhodedov, 2015) addresses the multi-item, multi-bidder auction design challenge by finding approximate optimal solutions. There are three main approaches in this field: the RegretNet-like approach (Dütting et al., 2024; Curry et al., 2020; Peri et al., 2021; Rahme et al., 2021; Duan et al., 2022; Ivanov et al., 2022) constructing IC constraints; the affine maximizer auctions (AMA) (Roberts, 1979) and related methods (Likhodedov & Sandholm, 2004; Guo et al., 2017; Curry et al., 2022; Duan et al., 2023) modifying allocation with weights for higher revenue; and the approach characterizing utility functions for IC and strategy-proofness (Dütting et al., 2024; Shen et al., 2019; Wang et al., 2024). Automated mechanism design is increasingly used in online advertising auctions (Zhang et al., 2021; Liu et al., 2021; Liao et al., 2022).

(Ma et al., 2024) proposed the joint advertisement setting, a Revised VCG mechanism for revenue improvement, and

JAMA (a menu-based AMA approach (Curry et al., 2022)) to further increase revenue. Our approach is closer to JRegNet (Zhang et al., 2024), a RegretNet-like method for joint ads. JRegNet encodes bipartite relationships in neural networks and transforms allocation to payment functions, outperforming VCG in small-bidder bipartite settings. But both JAMA and JRegNet have limitations. JAMA can't adapt to complex bipartite relationships in joint ads, and JRegNet has poor generalization in large, complex scenarios. (Aggarwal et al., 2024) proposed learning algorithms for repeated joint ads to minimize regret and maximize revenue in different environments, not focusing on single-round ad revenue.

## 3. Preliminaries

In this section, we mainly introduce the basic setting of joint advertising system.

In the context of joint advertisements, the sets of retailers $R$ and suppliers $S$ are mutually exclusive, with $R \cap S = \emptyset$. When a user submits a query, the advertising system retrieves $n$ advertisements represented by the set $E = \{e_1, \ldots, e_n\}$, where each advertisement links a retailer $r \in R$ with a supplier $s \in S$. Simultaneously, there are $m$ slots $M = \{1, \ldots, m\}$ available for displaying advertisements. The click-through rate (CTR) for the $k$-th slot is denoted as $\lambda_k$, where $k \in M$, and we assume that the CTRs are ordered such that $0 \leq \lambda_m \leq \cdots \leq \lambda_k \leq \cdots \leq \lambda_1 \leq 1$. We represent these CTRs using a vector, $\boldsymbol{\lambda} = (\lambda_1, \ldots, \lambda_m)$. The relationship between $R$ and $S$ is modeled as a bipartite graph $G = (R, S, E)$, with edges $E \subseteq R \times S$ representing advertisements.

The joint valuation distribution domain is defined as $V = V^R \times V^S$, where $V^R$ represents the domain of all possible retailer valuation distributions, with $v^R = (v_r)_{r \in R}$, and $V^S$ is the domain of all possible supplier valuation distributions, with $v^S = (v_s)_{s \in S}$. Retailer valuations $v_r$ and supplier valuations $v_s$ are independently drawn from their respective cumulative distribution functions $F_r$ and $F_s$, with probability density functions $f_r(v_r)$ and $f_s(v_s)$. The domain excluding retailer $r$ is $V_{-r} = V_{-r}^R \times V^S$, where $V_{-r}^R = \prod_{r' \in R \setminus \{r\}} V_{r'}^R$, and the domain excluding supplier $s$ is $V_{-s} = V^R \times V_{-s}^S$, where $V_{-s}^S = \prod_{s' \in S \setminus \{s\}} V_{s'}^S$.

The auctioneer knows the distribution $F = (F_i)_{i \in R \cup S}$, but not the realized valuation profile $\mathbf{v} = (v_i)_{i \in R \cup S}$. Bidders report their valuations as bids $\mathbf{b} = (b_i)_{i \in R \cup S}$, where $b_r \in V_r$ for retailer $r$ and $b_s \in V_s$ for supplier $s$. The auctioneer's personal value for each slot, if unallocated, is denoted by $v_0$.

Joint advertisements involving retailers or suppliers are defined as $E_r = \{(r^*, s) \in E \mid r^* = r\}$ for retailer $r$ and $E_s = \{(r, s^*) \in E \mid s^* = s\}$ for supplier $s$. Excluded joint advertisements are denoted as $E_{-r}$ and $E_{-s}$, representing bundles that exclude a specific retailer $r$ or supplier $s$, respectively.

**Definition 3.1** (Joint Auction Mechanism)**.** The Joint Auction Mechanism is represented by a pair of rules $\mathcal{M} = (x, p)$. The allocation rule of bundle $e$ is denoted by $x^e : V \to 2^M$, and the payment rule of bidder $i$ for bundle $e$ is denoted by $p_i^e : V \to \mathbb{R}_{\geq 0}$. The joint auction mechanism $\mathcal{M} = (x, p)$ is then defined with the following components:

1. **Allocation and Payment Rules**: For each bundle $e = (r, s) \in E$, the allocation and payment are shared between the participating bidders $r \in R$ and $s \in S$ as follows:

$$x^e(v) = x_r^e(v) = x_s^e(v), \quad \forall e = (r, s) \in E,$$

$$p^e(v) = p_r^e(v) + p_s^e(v), \quad \forall e = (r, s) \in E.$$

The total allocation and payment for a participant $i \in R \cup S$ are:

$$x_i(v) = \sum_{e \in E_i} x^e(v), \quad p_i(v) = \sum_{e \in E_i} p_i^e(v). \quad (1)$$

Since each slot can only be allocated to a single bundle, the allocation must satisfy the constraint:

$$\sum_{e \in E} x^e(v) \leq \mathbf{1}_m. \quad (2)$$

2. **Expected Quasilinear Utility**: The expected utility for a participant $i \in R \cup S$ is given by:

$$U_i(v_i, v_i') = \mathbb{E}_{v_{-i} \sim V_{-i}} \left[ v_i x_i(v_i', v_{-i}) \boldsymbol{\lambda}^T - p_i(v_i', v_{-i}) \right].$$

3. **Incentive Compatibility**: A mechanism satisfies IC if:

$$U_i(v_i, v_i) \geq U_i(v_i, v_i'), \quad \forall i \in R \cup S, \ v_i, v_i' \in V_i.$$

4. **Individual Rationality**: A mechanism satisfies IR if:

$$U_i(v_i, v_i) \geq 0, \quad \forall i \in R \cup S, \ v_i \in V_i.$$

5. **Expected Revenue**: The expected total revenue of the mechanism is:

$$U_0 = \mathbb{E}_{\mathbf{v} \sim V} \left[ v_0 \left( \mathbf{1} - \sum_{e \in E} x^e(\mathbf{v}) \right) \boldsymbol{\lambda}^T + \sum_{e \in E} p^e(\mathbf{v}) \right].$$

## 4. Optimal Joint Auction Design with Single Slot

We say that a mechanism $\mathcal{M}$ is feasible if and only if the mechanism $\mathcal{M}$ satisfies the conditions of IR and IC. For the single-item environment, the most well-known theory for this case is Myerson's Lemma (Myerson, 1981), which provides a necessary and sufficient condition for the feasibility of a mechanism.

**Lemma 4.1** (Myerson's Lemma). *In the single-item setting, a mechanism $(x, p)$ is feasible if and only if the following conditions hold:*

(a) *An allocation rule $x$ is monotonically non-decreasing.*

(b) *If $x$ is value-monotonic, then there is a unique payment rule $p$ for which $(x, p)$ is IR and IC and the winner pays her critical bid and the losers pay zero.*

Extending this framework to a single-slot joint advertisement scenario, the feasibility conditions remain consistent with traditional auction mechanisms. The primary difference lies in the allocation and payment rules, which are adapted to account for joint advertisement characteristics. Myerson-like mechanisms can thus be applied to the joint advertisement setting through appropriate modifications.

In this paper, we consider distributions that satisfy the *regular* condition, which is crucial in Myerson's auction theory:

**Definition 4.2.** A distribution $V$ is termed **regular** if its associated *virtual value function*

$$c(v) = v - \frac{1 - F(v)}{f(v)}$$

is monotonically increasing function of $v$ in the support of $V$.

To formulate these modifications, we first introduce the concept of bundles and neighbors, which are critical for defining the allocation and payment rules in the optimal joint auction mechanism. The neighbors of a node are defined as $N(r) = \{s \in S \mid (r, s) \in E\}$ for retailer $r$ and $N(s) = \{r \in R \mid (r, s) \in E\}$ for supplier $s$. These structures allow us to define the virtual value functions for retailers and suppliers as $c_r(v_r) = v_r - \frac{1 - F_r(v_r)}{f_r(v_r)}$ and $c_s(v_s) = v_s - \frac{1 - F_s(v_s)}{f_s(v_s)}$, respectively. Using these virtual values, we identify the maximum adjacent nodes, where $s_r^M = \arg\max_{s \in N(r)} c_s(v_s)$ for retailer $r$ and $r_s^M = \arg\max_{r \in N(s)} c_r(v_r)$ for supplier $s$. The corresponding bundles are $e_r^M = (r, s_r^M)$ and $e_s^M = (r_s^M, s)$, and the virtual value of a bundle $e = (r, s)$ is defined as the sum of the virtual values of its nodes: $c^e(v_r, v_s) = c_r(v_r) + c_s(v_s)$. These bundle and neighbor definitions serve as the foundation for the allocation and payment rules in the optimal mechanism.

Building on these structures, we establish the necessary and sufficient conditions for an optimal joint auction mechanism in the single-slot scenario. The optimal mechanism aims to achieve both feasibility (satisfying IR and IC) and revenue maximization.

**Theorem 4.3.** *For the single-slot joint advertisement with regular bidders, A deterministic joint auction mechanism $\mathcal{M}$ is optimal if and only if for all $i \in R \cup S$,*

(i) *Step Function:*

$$x_i^{\mathcal{M}}(v_i, v_{-i}) = \begin{cases} 1 & \text{if } v_i > \hat{v}_i(v_{-i}) \\ 0 & \text{otherwise} \end{cases}.$$

(ii) *Critical Value:*

$$p_i^{\mathcal{M}}(v_i, v_{-i}) = \begin{cases} \hat{v}_i(v_{-i}) & \text{if } v_i > \hat{v}_i(v_{-i}) \\ 0 & \text{otherwise} \end{cases}.$$

*where the critical value $\hat{v}_i(v_{-i})$ is defined as follows:*

• *For $r \in R$, the critical value $\hat{v}_r(v_{-r})$ is:*

$$\hat{v}_r(v_{-r}) = \inf\{b_r \mid c_r^{e^M}(b_r, v_{s_r^M}) \geq v_0 \wedge$$
$$c_r^{e^M}(b_r, v_{s_r^M}) \geq c^{\hat{e}}(v_{\hat{r}}, v_{\hat{s}}), \forall \hat{e} \in E_{-r}\}.$$

• *For $s \in S$, the critical value $\hat{v}_s(v_{-s})$ is defined similarly:*

$$\hat{v}_s(v_{-s}) = \inf\{b_s \mid c_s^{e^M}(v_{r_s^M}, b_s) \geq v_0 \wedge$$
$$c_s^{e^M}(v_{r_s^M}, b_s) \geq c^{\hat{e}}(v_{\hat{r}}, v_{\hat{s}}), \forall \hat{e} \in E_{-s}\}.$$

See Appendix B for detailed proofs.

# 5. Optimal Joint Auction Design as Learning Problem

In multi-slot joint advertisement, bids for bundles are jointly determined by two bidders, making them interdependent and challenging to handle. To address this, we propose **BundleNet**, a novel neural network architecture with a corresponding learning methodology for optimal mechanism design. Section 5.1 introduces a bundle-based IC constraint, Section 5.2 details BundleNet's architecture, and Section 5.3 elaborates on its loss function and training process.

## 5.1. Differentiable Approximation Approach for Joint Auction Design

In the domain of automated mechanism design, particularly within the class of RegretNet-based approaches (Dütting et al., 2024; Duan et al., 2022; Ivanov et al., 2022; Zhang et al., 2024), the optimization problem typically involves balancing two conflicting objectives: maximizing revenue and minimizing regret. The trade-off between these objectives is controlled by hyperparameters, such as the initial values and schedules of the Lagrangian multipliers.

In the context of joint advertisements, (Zhang et al., 2024) extended this framework by introducing IC constraints for all participants, including both retailers and suppliers. Their

model is formalized as:

$$\min_{w \in \mathbb{R}^d} \quad -\mathbb{E}_{v \sim V} \left[ \sum_{i \in R \cup S} p_i(v; w) \right]$$

$$\text{s.t.} \quad rgt_i(w) = 0, \quad \forall i \in R \cup S,$$

where $w$ represents the parameters of the neural network used for the automated mechanism. The regret function $rgt_i(w)$ is defined as:

$$rgt_i(w) = \mathbb{E}_{v \sim V} \left[ \max_{v_i' \in V_i} u_i(v_i; (v_i', v_{-i}); w) - u_i(v_i; (v_i, v_{-i}); w) \right].$$

In joint advertisement scenarios, the bidding strategies of bidders have different impacts on the mechanism, as their degrees in the bipartite graph are not identical. However, from the perspective of bundles in joint advertisements, we observe that the properties of each bundle are generally similar. Each bundle contains bids from both retailers and suppliers. Therefore, we propose redefining the IC constraints by constructing them for bundles instead of individual bidders. Although it may seem that a bundle, consisting of both retailers and suppliers, could be modeled as a two-parameter auction problem, the bundle itself does not have a strategy: It depends on the bidding strategies of the connected retailer and supplier. As a result, directly defining an IC constraint for bundles is highly challenging. To address this, we propose a method to expand the feasible domain, which derives an IC constraint from the bundle's perspective. The key feature of this new constraint is that it encloses the feasible region defined by the original IC constraints. By forcing the new constraint to approach zero, the original IC constraints also approach zero, ultimately ensuring the incentive compatibility of the entire mechanism. We define the ex-post regret for bundles as follows:

$$rgt^e(w) =$$

$$\mathbb{E}_{v \sim V} \left[ \max_{v_r' \in V_r} \{ u_r^e(v_r, (v_r', v_{-r}); w) - u_r^e(v_r, (v_r, v_{-r}); w) \} \right.$$

$$\left. + \max_{v_s' \in V_s} \{ u_s^e(v_s, (v_s', v_{-s}); w) - u_s^e(v_s, (v_s, v_{-s}); w) \} \right].$$

Lemma 5.1 demonstrates the relationship between the original constraints and the newly introduced constraints:

**Lemma 5.1.** *In a joint advertisement, the sum of the IC constraints for bundles is always greater than or equal to the sum of the IC constraints for individual bidders, as follows:*

$$\sum_{i \in R \cup S} rgt_i(w) \leq \sum_{e \in E} rgt^e(w).$$

The detailed proof is provided in Appendix C. IC constraints are designed to penalize situations where the utility from

misreporting exceeds that of truthful reporting. When the utility from misreporting is less than the utility from truthful bidding, the IC constraint becomes inactive. Therefore, $rgt_i(w) \geq 0$. When the sum of penalties imposed by the bundle IC constraints approaches zero, the IC penalties for all bidders also approach zero. With Lemma 5.1 guaranteeing the desired properties, we reformulate the optimal joint advertisement design as the following constrained optimization problem:

$$\min_{w \in \mathbb{R}^d} \quad -\mathbb{E}_{v \sim V} \left[ \sum_{e \in E} p^e(v; w) \right] \tag{3}$$

$$\text{s.t.} \quad rgt^e(w) = 0, \quad \forall e \in E.$$

To solve this, we aim to use sampling-based methods to learn the mechanism $\mathcal{M} = (x, p)$. Given a sample $S$ of $L$ valuation profiles drawn from a distribution $F$, we estimate the ex-post regret for each bundle as:

$$\widehat{rgt}^e(w) =$$

$$\frac{1}{L} \sum_{\ell=1}^{L} \left[ \max_{v_r' \in V_r} \left\{ u_r^e\big(v_r^{(\ell)}, (v_r', v_{-r}^{(\ell)}); w\big) - u_r^e\big(v_r^{(\ell)}, v^{(\ell)}; w\big) \right\} \right.$$

$$\left. + \max_{v_s' \in V_s} \left\{ u_s^e\big(v_s^{(\ell)}, (v_s', v_{-s}^{(\ell)}); w\big) - u_s^e\big(v_s^{(\ell)}, v^{(\ell)}; w\big) \right\} \right].$$

Finally, we reformulate the original optimization problem as minimizing the empirical loss of negated revenue, subject to the constraint that the empirical ex-post regret for all bundles is zero:

$$\min_{w \in \mathbb{R}^d} \quad -\frac{1}{L} \sum_{\ell=0}^{L} \sum_{e \in E} p^e(v^{(\ell)}; w) \tag{4}$$

$$\text{s.t.} \quad \widehat{rgt}^e(w) = 0, \quad \forall e \in E.$$

Additionally, we utilize a neural network architecture to ensure that the mechanism satisfies the conditions of IR in Sec 5.2.

### 5.2. Neural Network Architecture

This section introduces our neural network-based approach for mechanism design in joint advertisements. Our architecture comprises two primary components: **Allocation Network** and **Payment Network** . As illustrated in Figure 2, we employ a graph-based approach to model the interactions between retailers and suppliers effectively. This process, termed **Graph Feature Fusion**, involves aggregating node features from a bipartite graph into edge features that capture the combined characteristics of retailer-supplier pairs (bundles).

Each node in the bipartite graph $G = (R, S, E)$ is associated with a feature vector representing the bidder's cost per click

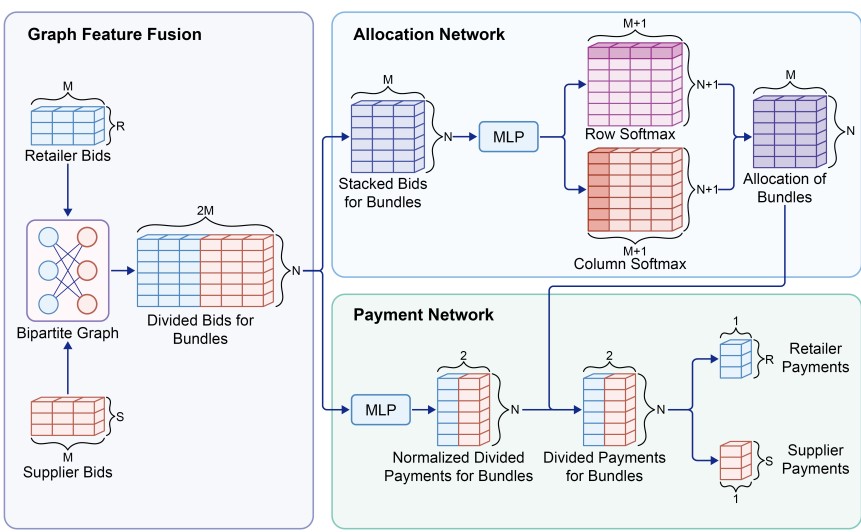

*Figure 2.* The neural network architecture BundleNet. Details are shown in Sec 5.2.

(CPC), defined as $X_r = b_r \boldsymbol{\lambda} \in \mathbb{R}_{\geq 0}^m$ for retailer $r$ or $X_s = b_s \boldsymbol{\lambda} \in \mathbb{R}_{\geq 0}^m$ for supplier $s$.

To capture the combined influence of retailers and suppliers on each bundle, we aggregate their node features into edge features, which we refer to as **Divided Bids for the Bundle**. This process is mathematically formalized as follows:

$$DB^e = [X_r, X_s] \in \mathbb{R}_{\geq 0}^{2m}, \quad \forall e = (r, s) \in E,$$

where $e = (r, s)$ represents an edge between a retailer $r$ and a supplier $s$. We denote $DB^E$ as the features of all edges in the bipartite graph.

By summing up the aggregated edge features, we derive the **Stacked Bids for the Bundle**, which is defined as

$$SB^e = X_r + X_s \in \mathbb{R}_{\geq 0}^m, \quad \forall e = (r, s) \in E.$$

Similarly, we use $SB^E$ to represent the aggregated features of all edges in the bipartite graph.

In the **Allocation Network**, similar to RegretNet, we utilize a doubly stochastic matrix approach to ensure that each bundle is allocated to exactly one slot and each slot is assigned to only one bundle, reflecting the joint advertisement scenario. A key property of doubly stochastic matrices is encapsulated in the following lemma:

**Lemma 5.2** ((Dütting et al., 2024)). *The matrix $\phi^{DS}(x, x')$ is doubly stochastic $\forall x, x' \in \mathbb{R}^{nm}$. For any doubly stochastic matrix $a \in [0, 1]^{nm}$, there exist $x, x' \in \mathbb{R}^{nm}$, for which $a = \phi^{DS}(x, x')$.*

$$a_{ij} = \phi_{ij}^{DS}(x, x') = \min \left\{ \frac{e^{x_{ij}}}{\sum_{k=1}^{n+1} e^{x_{kj}}}, \frac{e^{x'_{ij}}}{\sum_{k=1}^{m+1} e^{x'_{ik}}} \right\}.$$

The aggregated edge features $SB^E$, serve as the input to a multi-layer perceptron (MLP), producing an intermediate output denoted as $Y \in \mathbb{R}^{d_y}$, where $d_y$ represents the dimensionality of the output feature space:

$$Y = \text{MLP}(SB^E) \in \mathbb{R}^{d_y}.$$

The output $Y$ contains values that reflect the potential allocation results for each bundle across available slots. To normalize these values, we apply two distinct transformations using matrices $W_{\text{row}} \in \mathbb{R}^{d_y \times (n+1)(m+1)}$ and $W_{\text{col}} \in \mathbb{R}^{d_y \times (n+1)(m+1)}$.

The first transformation involves multiplying $Y$ by $W_{\text{row}}$, resulting in a matrix that captures the allocation potential for each slot:

$$MR = YW_{\text{row}} \in \mathbb{R}^{(n+1)(m+1)}.$$

Next, we apply the row-wise softmax function to $R$. The softmax function transforms the raw scores into a probability distribution across the bundles for each slot.

$$DR_{ij} = \frac{\exp(MR_{ij})}{\sum_{k=1}^{m+1} \exp(MR_{ik})}, \quad \forall i \in \{1, \ldots, n+1\}.$$

Similarly, we perform a second transformation by multiplying $Y$ by $W_{\text{col}}$:

$$MC = YW_{\text{col}} \in \mathbb{R}^{(n+1)(m+1)}.$$

We then apply the column-wise softmax function to $C$. For each column $j$ in $C$, the softmax is defined as:

$$DC_{ij} = \frac{\exp(MC_{ij})}{\sum_{k=1}^{n} \exp(MC_{k,j})}, \quad \forall j \in \{1, \ldots, m+1\}.$$

| Alg. | Setting | | | | | | | |
|---|---|---|---|---|---|---|---|---|
| | $U_2$ | $U_3$ | $U_4$ | $U_5$ | $E_2$ | $E_3$ | $E_4$ | $E_5$ |
| **Ours** | | | | | | | | |
| BundleNet | **0.5286** | **0.6681** | **0.7805** | **0.8802** | **0.4248** | **0.5460** | **0.6354** | **0.7215** |
| IC Violation | 0.0006 | < 0.001 | < 0.001 | < 0.001 | < 0.001 | < 0.001 | < 0.001 | < 0.001 |
| IC Baselines | | | | | | | | |
| RVCG | 0.3811 | 0.6003 | 0.7455 | 0.8607 | 0.2820 | 0.4649 | 0.5927 | 0.7041 |
| Optimal | 0.5247 | 0.6705 | 0.7826 | 0.8819 | 0.4249 | 0.5479 | 0.6470 | 0.7376 |
| Baselines with IC Violation | | | | | | | | |
| JRegNet | 0.5622 | 0.7287 | 0.7791 | 0.7882 | 0.4727 | 0.5892 | 0.6306 | 0.6943 |
| IC Violation | 0.00054 | < 0.001 | < 0.001 | < 0.001 | < 0.001 | < 0.001 | < 0.001 | < 0.001 |

*Table 1.* The experimental results compare BundleNet, JRegNet, Revised VCG, and the Optimal Mechanism as the number of bundles increases under different settings in the single-slot scenario with CTR $\lambda = (1)$. The notations $U_2, U_3, U_4, U_5$ represent cases where the number of bundles is 2, 3, 4 and 5, respectively, under the uniform distribution $U(0, 1)$. Similarly, $E_2, E_3, E_4, E_5$ correspond to scenarios with 2, 3, 4 and 5 bundles under the truncated exponential distribution $E(2)$. In this table, we use bold to indicate the method among BundleNet, JRegNet, and RVCG that is closest to the optimal mechanism, rather than the one with the highest revenue.

Finally, to ensure consistency in the allocation, we take the element-wise minimum of the two resulting matrices to obtain the final allocation matrix $A$, where:

$$A_{ij} = \min(DR_{ij}, DC_{ij}), \quad \forall i \in \{1, \ldots, n\}, j \in \{1, \ldots, m\}.$$

In the **Payment Network**, to ensure that the auction satisfies ex-post IR, we generate normalized payments for each bundle corresponding to the retailer and supplier, denoted as $\tilde{p}_r^i$ and $\tilde{p}_s^i \in [0, 1]$. We employ an MLP with a Sigmoid activation function in the final layer to represent this mapping:

$$\tilde{p} = \text{Sigmoid}(\text{MLP}(DB^E)) \in \mathbb{R}^{2n}.$$

Subsequently, we derive the corresponding payments as follows:

$$p_r^{e_i} = \tilde{p}_r^{e_i} \sum_{j=1}^m A_{ij} X_r[j],$$

$$p_s^{e_i} = \tilde{p}_s^{e_i} \sum_{j=1}^m A_{ij} X_s[j].$$

Finally, we can determine the payment for each participating bidder in the advertising auction using the equations defined in Equation (1).

### 5.3. Optimization and Training

We optimize the constrained objective (4) by introducing the augmented Lagrangian method. Our loss function is formulated as follows:

$$\mathcal{L}_\rho(w; \mu) =$$
$$-\frac{1}{L} \sum_{\ell=1}^L \sum_{e \in E} p^e(v^{(\ell)}) + \sum_{e \in E} \mu_e \widehat{rgt}^e(w) + \frac{\rho}{2} \sum_{e \in E} \left(\widehat{rgt}^e(w)\right)^2.$$

where $w$ represents the neural network parameters, $\lambda_e$ represents the Lagrangian multipliers associated with the constraints, while $\rho$ is a hyper-parameter controlling the weight of the quadratic penalty term. During optimization, we utilize the Adam optimizer to update our parameters $w$ as well as the misreports $v_r'^{(\ell)}$ and $v_s'^{(\ell)}$ in turn, i.e., we update $w^{new} \in \arg\min_w \mathcal{L}_\rho(w^{old}, \mu^{old})$ and update $\mu_e^{new} = \mu_e^{old} + \rho \cdot rgt^e(w^{new}), \forall e \in E$. The detailed algorithmic specifications can be found in Algorithm 1.

## 6. Experiments

In this chapter, we present **empirical experiments** to demonstrate the effectiveness of BundleNet. All experiments are conducted on a Linux machine equipped with NVIDIA Graphics Processing Unit cores.

**Baseline methods:** We compare BundleNet with the following baselines:

- **Optimal Joint Auction Mechanism**, a Myerson-like method for optimal mechanism design in joint advertising auctions with a single slot (See Sec. 4).

- **VCG** (Vickrey, 1961; Clarke, 1971; Groves, 1973), a classic mechanism satisfying DSIC and IR. In our experiments, we apply the RVCG mechanism (Ma et al., 2024) to the joint advertising.

- **JRegNet** (Zhang et al., 2024), a neural network architecture for near DSIC mechanism design in the joint advertising auction settings which can achieve the near-optimal revenue.

**Evaluation:** We generate training and test data from different distributions. The training set consists of 204,800

| Alg. | Setting | | | | | | | |
|---|---|---|---|---|---|---|---|---|
| | $U_{5\times5}$ | $U_{6\times5}$ | $U_{7\times5}$ | $U_{8\times5}$ | $U_{9\times5}$ | $U_{10\times5}$ | $U_{11\times5}$ | $U_{12\times5}$ |
| **Ours** | | | | | | | | |
| BundleNet | **1.4982** | **1.7162** | **1.9233** | **2.0890** | **2.2210** | **2.4047** | **2.5649** | **2.6495** |
| IC Violation | $< 0.001$ | $< 0.001$ | $< 0.001$ | $< 0.001$ | $< 0.001$ | $< 0.001$ | $< 0.001$ | $< 0.001$ |
| IC Baseline | | | | | | | | |
| RVCG | 0.8420 | 1.2854 | 1.6142 | 1.8831 | 2.0991 | 2.2800 | 2.4564 | 2.5868 |
| Baselines with IC Violation | | | | | | | | |
| JRegNet | 1.4972 | 1.6849 | 1.8244 | 1.9350 | 1.9804 | 1.9622 | 1.9763 | 1.9973 |
| IC Violation | $< 0.001$ | $< 0.001$ | $< 0.001$ | $< 0.001$ | $< 0.001$ | $< 0.001$ | $< 0.001$ | $< 0.001$ |

*Table 2.* The experimental results of BundleNet, JRegNet, RVCG as the number of bundles increases under different settings in the multi-slots scenario. Similar to those of Table 1, the notation $U_{5\times5}, \cdots, U_{12\times5}$ represent the settings where the number of bundles varies from 5 to 12, while letting the CTRs of these 5 slots as (1, 0.8, 0.6, 0.4, 0.2).

samples, while the test set contains 20,480 samples. To assess the performance of each method, we utilize the average empirical ex-post regret of the mechanism: $\widehat{rgt} := \frac{1}{2n}\sum_{i\in R\cup S}\widehat{rgt}_i$. Since in real-world scenarios, $v_0 = 0$, we also consider empirical revenue: rev $:= \frac{1}{L}\sum_{\ell=1}^{L}\sum_{e\in E}p^e(v^{(\ell)})$. In all synthetic data experiments, the joint relationship matrix between stores and brands is randomly generated for each search request sample. We evaluate the revenue performance of BundleNet, JRegNet, and RVCG across different distributions.

### 6.1. Experimental Results for the Single Slot

The experimental setup considers the optimal joint auction mechanism in a single-slot setting, evaluated under three different probability distributions commonly studied in the literature: **Setting U:** The uniform distribution $U(0,1)$. **Setting E:** The truncated exponential distribution $Exp(2)$ over the interval $(0,1)$. **Setting N:** The truncated normal distribution $N(0.5, 0.1)$ over the interval $(0,1)$. For each distribution, we examine auction scenarios with a varying number of bundles, specifically ranging from 2 to 5, with a single-slot CTR of 1.

The experimental results, reported in Table 1 and Table 3 , illustrate the performance of different auction mechanisms under these settings. The experimental results demonstrate that BundleNet consistently approximates the optimal mechanism across various settings, whereas JRegNet does not always exhibit such proximity. This indicates that, although both methods are based on RegretNet, BundleNet improves upon it by modifying the neural network architecture and optimization approach, enabling it to better learn the optimal mechanism. We provide a visual analysis in the Appendix E. The allocation results of BundleNet are much closer to the optimal mechanism, while the allocation results of JRegNet are significantly different from the optimal mechanism.

### 6.2. Experimental Results for the Multi-Slot

We have also conducted extensive experiments to evaluate the performance of BundleNet under the multi-slot scenario from experiments on real-world dataset (Zhang et al., 2024).

Concretely, we explore different variations of this setting, where 10 bundles compete for 5 slots. We fix the number of slots and vary the number of bundles to evaluate the performance of BundleNet in comparison with baseline mechanisms. The bidders' value profiles are sampled from three different probability distributions: **Setting U**, where values follow the uniform distribution $U(0,1)$; **Setting LN**, where values follow the truncated lognormal distribution $LN(0.1, 1.44)$ over the interval $(0,1)$; **Setting N**, where values follow the truncated normal distribution $N(0.5, 0.1)$ over the interval $(0,1)$. For each distribution, we assess different mechanisms by varying the number of bundles from 5 to 12, while keeping the CTRs of the 5 slots fixed at (1, 0.8, 0.6, 0.4, 0.2).

Regarding the experimental results for **Setting U**, as shown in Table 2, we conclude that BundleNet achieves higher revenue compared to other baseline mechanisms. BundleNet consistently outperforms VCG across all scenarios, whereas JRegNet surpasses VCG only in some cases. The experimental results for **Setting LN** and **Setting N**, provided in Appendix D.2, lead to similar conclusions.

## 7. Conclusion

We propose two solutions for the optimal mechanism design in joint advertisement. The first solution in the single-slot scenario, is a Myerson-based approach for the single-slot scenario and an automated RegretNet-inspired method for the multi-slot case. Empirical results show that BundleNet learns a Myerson-like mechanism in the single-slot setting and finds a near-optimal solution outperforming baselines in the multi-slot setting.

## Acknowledgments

This work was supported by National Natural Science Foundation of China No. 62472428, Public Computing Cloud Renmin University of China and fund for building world-class universities (disciplines) of Renmin University of China.

## Impact Statement

This paper presents work whose goal is to advance the field of Machine Learning. There are many potential societal consequences of our work, none which we feel must be specifically highlighted here.

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

## A. Optimization and Training Procedures

---

**Algorithm 1** BundleNet Training

---

**Input:** Minibatches $\mathcal{B}_1, \ldots, \mathcal{B}_T$ of size $C$
**Parameters:** $\forall t, \rho_t > 0, \gamma > 0, \eta > 0, \Gamma \in \mathbb{N}, K \in \mathbb{N}$
**Initialize:** $w^0 \in \mathbb{R}^d, \lambda^0 \in \mathbb{R}^m$
**for** $t = 0$ **to** $T$ **do**
    Receive $\mathcal{B}_t = \{G^{(1)}, \ldots, G^{(B)}\}$
    Initialize $v_r^{'(\ell)} \in V_r, v_s^{'(\ell)} \in V_s, \forall \ell \in [B], r \in R, s \in S$
    **for** $r = 0$ **to** $\Gamma$ **do**
        **for** $\ell = 1$ **to** $B$ **do**
            $\forall \ell \in [C], i \in R \cup S :$
            $v_i^{'(\ell)} \leftarrow v_i^{'(\ell)} + \gamma \nabla_{v_i'} [u_i^w(v_i^{(\ell)}; (v_i', v_{-i}^{(\ell)}))]\big|_{v_i' = v_i^{'(\ell)}}$
        **end for**
    **end for**
    Compute Lagrangian gradient and update $w^t$
    $w^{t+1} \leftarrow w^t - \eta \nabla_w \mathcal{L}_{\rho_t}(w^t, \mu^t)$
    **if** $t$ is a multiple of $H$ **then**
        $\mu_e^{t+1} \leftarrow \mu_e^t + \rho_t \widetilde{rgt}^e(w^{t+1}), \forall e \in E$
    **else**
        $\boldsymbol{\lambda}^{t+1} \leftarrow \boldsymbol{\lambda}^t$
    **end if**
**end for**

---

## B. Proof of Theorem 4.3

The joint auction with single slot remains a single-parameter auction, with the primary distinction from traditional auctions lying in the allocation and payment rules. As a result, the necessary and sufficient conditions for a feasible auction, as established in Myerson's work (Myerson, 1981), still hold in this setting. Before presenting the proof, we first introduce an assumption: suppose $a_i$ and $b_i$ are the lower and upper bounds of the domain of the probability density function $f_i$ for bidder $i$. That is, $v_i \in [a_i, b_i]$. Given a joint auction mechanism, we define $Q_i(v_i)$ as:

$$Q_i(v_i) = \mathbb{E}_{\mathbf{v}_{-i} \sim F_{-i}}[x_i(v_i)], \quad \forall i \in R \cup S,$$

where represents the expected probability that bidder $i$ wins, given their value estimate $v_i$.

**Lemma B.1** ((Myerson, 1981))**.** *In the single-slot setting, a joint auction mechanism $(x, p)$ is feasible if and only if the following conditions hold:*

(a) *If $v_i^* \leq v_i$, then $Q_i(v_i^*) \leq Q_i(v_i), \quad \forall i \in R \cup S, v_i, v_i^* \in V_i$.*

(b) *$U_i(v_i, v_i) = U_i(a_i, a_i) + \int_{a_i}^{v_i} Q(z)dz, \quad \forall i \in R \cup S.$*

(c) *$U_i(v_i, v_i) \geq 0, \quad \forall i \in R \cup S.$*

(d) *$\sum_{e \in E} x^e(\mathbf{v}) \leq 1, \quad x^e(\mathbf{v}) \geq 0$*

$\mathcal{M} = (x, p)$ is an optimal joint auction mechanism if and only if it maximizes $U_0$ while ensuring the feasibility of the auction. We provide a simpler condition to characterize optimality.

**Lemma B.2.** *Suppose that the function $x : V \to 2^n$ maximizes the following objective:*

$$\max_{\mathcal{M}} \quad \int_V \sum_{e=(r,s) \in E} \left( v_r + v_s - \frac{1 - F_r(v_r)}{f_r(v_r)} - \frac{1 - F_s(v_s)}{f_s(v_s)} - v_0 \right) x^e(v)f(v)dv$$

$$s.t. \quad p_i(v) = v_i x_i(\mathbf{v}) - \int_{a_i}^{v_i} x_i(z, v_{-i})dz, \quad \forall i \in R \cup S$$

$$\text{If } v_i^* \leq v_i, \text{ then } Q_i(v_i^*) \leq Q_i(v_i), \quad \forall i \in R \cup S, v_i, v_i^* \in V_i,$$

$$\sum_{e \in E} x^e(\mathbf{v}) \leq 1, \quad x^e(\mathbf{v}) \geq 0$$

(5)

*Then, the pair $\mathcal{M} = (x, p)$ represents an optimal joint auction.*

*Proof.* To prove that the pair $\mathcal{M} = (x, p)$ represents an optimal joint auction, we analyze the expected revenue function $U_0$ and show that it maximizes the given objective function while satisfying feasibility and incentive compatibility constraints. The expected revenue $U_0$ is given by:

$$
\begin{aligned}
U_0 &= \mathbb{E}_{\mathbf{v} \sim F} \left[ v_0 \left( 1 - \sum_{e \in E} x^e(\mathbf{v}) \right) + \sum_{e \in E} p^e(\mathbf{v}) \right], \\
&= \int_V v_0 f(v) dv + \sum_{e \in E} \left( \int_T [p_r^e(v) - v_r x_r^e(v)] f(v) dv + \int_V [p_s^e(v) - v_s x_s^e(v)] f(v) dv \right) \\
&\quad + \sum_{e \in E} \int_V x^e(v)(v_s + v_r - v_0) f(v) dv
\end{aligned}
\tag{6}
$$

Using the conclusion of Lemma B.1, we can simplify the following equation.

$$
\begin{aligned}
\sum_{e \in E} \int_V [p_r^e(v) - v_r x_r^e(v)] f(v) dv &= \sum_{r \in R} \int_V [p_r(v) - v_r x_r(v)] f(v) dv \\
&= -\sum_{r \in R} \int_{a_r}^{b_r} [U_r(v_r, v_r)] f_r(v_r) dv_r \\
&= -\sum_{r \in R} \int_{a_r}^{b_r} [U_r(a_r, a_r) + \int_{a_r}^{v_r} Q_r(z) dz] f_r(v_r) dv_r \\
&= -\sum_{r \in R} [U_r(a_r, a_r) + \int_{a_r}^{b_r} \int_z^{b_r} Q_r(z) f_r(v_r) dv_r dz] \\
&= -\sum_{r \in R} [U_r(a_r, a_r) + \int_{a_r}^{b_r} (1 - F_r(z)) Q_r(z) dz] \\
&= -\sum_{r \in R} [U_r(a_r, a_r) + \int_{a_r}^{b_r} (1 - F_r(v_r)) x_r(v_r) f_{-r}(v_{-r}) dv] \\
&= -\sum_{e \in E} [U_r^e(a_r, a_r) + \int_{a_r}^{b_r} (1 - F_r(v_r)) x_r^e(v_r) f_{-r}(v_{-r}) dv]
\end{aligned}
\tag{7}
$$

Substituting Equation 7 into Equation 6 gives us:

$$
\begin{aligned}
U_0 &= \int_V v_0 f(v) dv + \sum_{e=(r,s) \in E} \int_V x^e(v) \left( v_s + v_r - \frac{1 - F_r(v_r)}{f_r(v_r)} - \frac{1 - F_s(v_s)}{f_s(v_s)} - v_0 \right) f(v) dv \\
&\quad + \sum_{r \in R} U_r(a_r, a_r) + \sum_{s \in S} U_s(a_s, a_s)
\end{aligned}
\tag{8}
$$

The first term in Equation 8 is a constant, while the third and fourth terms are always non-negative due to the IR property of the auction. When these terms are equal to zero, the objective function reaches its maximum value without affecting the value of the first and second terms. Thus, by setting the third and fourth terms to zero, we obtain:

$$
\begin{aligned}
p_r(v) &= v_r x_r(v) - \int_{a_r}^{v_r} x_r(z, v_{-r}) dz \\
p_s(v) &= v_s x_s(v) - \int_{a_s}^{v_s} x_s(z, v_{-s}) dz
\end{aligned}
\tag{9}
$$

$\square$

Next, we proceed with the proof of Theorem 4.3:

*Proof of Theorem 4.3.* In this paper, we consider distributions that are regular. Since distributions in the regular class ensure the monotonicity required by Lemma B.2, they guarantee the IC property of the joint auction. As a result, we can simplify the objective function as follows:

$$\sum_{e=(r,s)\in E} (c^e(v_r, v_s) - v_0)x^e(v)$$

This implicitly indicates that the slot will be allocated to the bundle with the highest virtual value. For a single-slot joint auction, each bidder's bidding strategy depends solely on the neighbor node with the highest virtual value. This is because any bundle formed with other neighbors has no chance of winning the auction. Therefore, for each bidder, their critical value is given by:

- For $r \in R$, the critical value $\hat{v}_r(v_{-r})$ is:

$$\hat{v}_r(v_{-r}) = \inf \left\{ b_r \mid c^{e_r^M}(b_r, v_{s_r^M}) \ge v_0, \text{ and } c^{e_r^M}(b_r, v_{s_r^M}) \ge c^{\hat{e}}(v_{\hat{r}}, v_{\hat{s}}), \forall \hat{e} = (\hat{r}, \hat{s}) \in E_{-r} \right\}$$

- For $s \in S$, the critical value $\hat{v}_s(v_{-s})$ is similarly:

$$\hat{v}_s(v_{-s}) = \inf\{b_s \mid c^{e_s^M}(v_{r_s^M}, b_s) \ge v_0, \text{ and } c^{e_s^M}(v_{r_s^M}, b_s) \ge c^{\hat{e}}(v_{\hat{r}}, v_{\hat{s}}), \forall \hat{e} = (\hat{r}, \hat{s}) \in E_{-s}\}$$

We take a node $r \in R$ as an example. The allocation of all bundles connected to $r$ is given by:

$$x^{e_r^M}(b_r, v_{-r}) = \begin{cases} 1 & \text{if } b_r \ge \hat{v}_r(v_{-r}) \\ 0 & \text{if } b_r < \hat{v}_r(v_{-r}) \end{cases}$$

$$x^{e^*}(b_r, v_{-r}) = 0, \quad \forall e^* \in E_r \setminus \{e_r^M\}$$

Thus, we can derive the allocation result for $r$ as follows:

$$x_r(b_r, v_{-r}) = x^{e_r^M}(b_r, v_{-r}) + \sum_{e^* \in E_r/\{e_r^M\}} x^{e^*}(b_r, v_{-r}) = \begin{cases} 1 & \text{if } b_r \ge \hat{v}_r(v_{-r}) \\ 0 & \text{if } b_r < \hat{v}_r(v_{-r}) \end{cases}$$

We can derive the final payment result by solving the payment formula based on the Equation 9. We integrate the allocation rule to determine the payment function. Thus, the final payment for bidder $r$ is given by:

$$p_r(b_r, v_{-r}) = \begin{cases} z_r(v_{-r}) & \text{if } b_r \ge \hat{v}_r(v_{-r}) \\ 0 & \text{if } b_r < \hat{v}_r(v_{-r}) \end{cases}$$

For a node $s \in S$, the conclusion follows similarly. The allocation and payment rules apply symmetrically due to the structure of the joint auction mechanism, where each participant's strategy depends on their highest virtual value neighbor.

Thus, the proof is complete.

□

## C. Proof of Lemma 5.1

*Proof.*

$$
\begin{aligned}
\sum_{i \in R \cup S} rgt_i &= \sum_{i \in R \cup S} \mathbb{E}_{v \sim F} \left[ \max_{v_i' \in V_i} u_i(v_i; (v_i', v_{-i}); w) - u_i(v_i; (v_i, v_{-i}); w) \right] \\
&= \sum_{r \in R} \mathbb{E}_{v \sim F} \left[ \max_{v_r' \in V_r} u_r(v_r; (v_r', v_{-r}); w) - u_r(v_r; (v_r, v_{-r}); w) \right] \\
&\quad + \sum_{s \in S} \mathbb{E}_{v \sim F} \left[ \max_{v_s' \in V_s} u_s(v_s; (v_s', v_{-s}); w) - u_s(v_s; (v_s, v_{-s}); w) \right] \\
&= \sum_{r \in R} \mathbb{E}_{v \sim F} \left[ \max_{v_r' \in V_r} \sum_{e \in E_r} u_r^e(v_r; (v_r', v_{-r}); w) - \sum_{e \in E_r} u_r^e(v_r; (v_r, v_{-r}); w) \right] \\
&\quad + \sum_{s \in S} \mathbb{E}_{v \sim F} \left[ \max_{v_s' \in V_s} \sum_{e \in E_s} u_s^e(v_s; (v_s', v_{-s}); w) - \sum_{e \in E_s} u_s^e(v_s; (v_s, v_{-s}); w) \right] \\
&\leq \sum_{e \in E} \mathbb{E}_{v \sim F} \left[ \max_{v_r' \in V_r} u_r^e(v_r; (v_r', v_{-r}); w) - u_r^e(v_r; (v_r, v_{-r}); w) \right] \\
&\quad + \sum_{e \in E} \mathbb{E}_{v \sim F} \left[ \max_{v_s' \in V_s} u_s^e(v_s; (v_s', v_{-s}); w) - u_s^e(v_s; (v_s, v_{-s}); w) \right] \\
&= \sum_{e = (r,s) \in E} rgt^e(w)
\end{aligned}
\tag{10}
$$

$\square$

## D. Additional Experiments

### D.1. Experimental Results Under Single-Slot Setting N

| Alg. | Setting | | | |
|---|---|---|---|---|
| | $N_2$ | $N_3$ | $N_4$ | $N_5$ |
| **Ours** | | | | |
| BundleNet | **0.7750** | **0.8692** | **0.9134** | **0.9561** |
| IC Violation | $< 0.001$ | $< 0.001$ | $< 0.001$ | $< 0.001$ |
| IC Baselines | | | | |
| RVCG | 0.5492 | 0.8114 | 0.8956 | 0.9444 |
| Optimal | 0.7789 | 0.8656 | 0.9188 | 0.9582 |
| Baselines with IC Violation | | | | |
| JRegNet | 0.8183 | 0.9091 | 0.8747 | 0.9117 |
| IC Violation | $< 0.001$ | $< 0.001$ | $< 0.001$ | $< 0.001$ |

*Table 3.* The experimental results compare BundleNet, JRegNet, Revised VCG, and the Optimal Mechanism as the number of bundles increases under different settings in the single-slot scenario with CTR $\lambda = (1)$. The notations $N_2, N_3, N_4, N_5$ represent cases where the number of bundles is 2, 3, 4 and 5, respectively, under the normal distribution $N(0.5, 0.1)$. In this table, we use bold to indicate the method among BundleNet, JRegNet, and RVCG that is closest to the optimal mechanism, rather than the one with the highest revenue.

## D.2. Experimental Results Under Multi-Slot Setting

| Alg. | Setting | | | | | | | |
| --- | --- | --- | --- | --- | --- | --- | --- | --- |
| | $LN_{5\times5}$ | $LN_{6\times5}$ | $LN_{7\times5}$ | $LN_{8\times5}$ | $LN_{9\times5}$ | $LN_{10\times5}$ | $LN_{11\times5}$ | $LN_{12\times5}$ |
| **Ours** | | | | | | | | |
| BundleNet | **2.6560** | **3.0093** | **3.4346** | **3.7031** | **3.9952** | **4.2366** | **4.4618** | **4.6038** |
| IC Violation | $< 0.001$ | $< 0.001$ | $< 0.001$ | $< 0.001$ | $< 0.001$ | $< 0.001$ | $< 0.001$ | $< 0.001$ |
| IC Baseline | | | | | | | | |
| RVCG | 1.4365 | 2.1764 | 2.7130 | 3.1579 | 3.5104 | 3.8332 | 4.1019 | 4.3449 |
| Baselines with IC Violation | | | | | | | | |
| JRegNet | 2.6020 | 2.9237 | 3.1845 | 3.4139 | 3.441 | 3.2535 | 3.2747 | 3.3848 |
| IC Violation | $< 0.001$ | $< 0.001$ | $< 0.001$ | $< 0.001$ | $< 0.001$ | $< 0.001$ | $< 0.001$ | $< 0.001$ |

*Table 4.* The experimental results of BundleNet, JRegNet, VCG as the number of bundles increases under different settings in the multi-slots scenario. Similar to those of 1, the notation $LN_{5\times5}, \cdots, LN_{12\times5}$ represent the settings where the number of bundles varies from 5 to 12 , respectively, under the truncated lognormal distribution LN (0.1, 1.44) over the interval (0, 1), while letting the CTRs of these 5 slots as (1, 0.8, 0.6, 0.4, 0.2). In most of the scenarios, BundleNet report the better performance of JRegNet.

| Alg. | Setting | | | | | | | |
| --- | --- | --- | --- | --- | --- | --- | --- | --- |
| | $N_{5\times5}$ | $N_{6\times5}$ | $N_{7\times5}$ | $N_{8\times5}$ | $N_{9\times5}$ | $N_{10\times5}$ | $N_{11\times5}$ | $N_{12\times5}$ |
| **Ours** | | | | | | | | |
| BundleNet | 2.1393 | **2.3690** | **2.4914** | **2.6287** | **2.7020** | **2.7916** | **2.8428** | **2.8841** |
| IC Violation | $< 0.001$ | $< 0.001$ | $< 0.001$ | $< 0.001$ | $< 0.001$ | $< 0.001$ | $< 0.001$ | $< 0.001$ |
| IC Baseline | | | | | | | | |
| RVCG | 1.3110 | 2.0132 | 2.3565 | 2.5343 | 2.6395 | 2.7228 | 2.7840 | 2.8383 |
| Baselines with IC Violation | | | | | | | | |
| JRegNet | **2.2071** | 2.3537 | 2.4814 | 2.4740 | 2.5185 | 2.4711 | 2.4082 | 2.3272 |
| IC Violation | $< 0.001$ | $< 0.001$ | $< 0.001$ | $< 0.001$ | $< 0.001$ | $< 0.001$ | $< 0.001$ | $< 0.001$ |

*Table 5.* The experimental results of BundleNet, JRegNet, VCG as the number of bundles increases under different settings in the multi-slots scenario. Similar to those of 1, the notation $N_{5\times5}, \cdots, N_{12\times5}$ represent the settings where the number of bundles varies from 5 to 12, respectively, under the truncated normal distribution N (0.5, 0.1) over the interval (0, 1), while letting the CTRs of these 5 slots as (1, 0.8, 0.6, 0.4, 0.2). In most of the scenarios, BundleNet report the better performance of JRegNet.

# E. Comparison of Visualized Allocation Results Under the Setting $U_2$

To further validate our hypothesis, we visualize the allocation rules of BundleNet and JRegNet to examine their differences. Since the bipartite graph structures in joint advertisement scenarios can be highly complex, we select the **Setting** $U_2$, which consists of only two types of heterogeneous bipartite graphs. Based on these structures, we design two different sets of experiments.

In the first experiment, we consider a bipartite graph where Set $R$ contains two nodes $(r_1, r_2)$, Set $S$ contains a single node $(s_1)$, and there are two bundles, $e_1 = (r_1, s_1)$ and $e_2 = (r_2, s_1)$. To observe the impact of bidding on the allocation outcome of $e_1$, we fix the bid for $s_1$ at 0, 0.25, 0.5, and 0.75, while varying the bids of the two nodes $r_1, r_2$.

In the second experiment, we extend the complexity by introducing a bipartite graph where set $R$ contains two nodes $(r_1, r_2)$, Set $S$ contains two nodes $(s_1, s_2)$, and there are two bundles, $e_1 = (r_1, s_1)$ and $e_2 = (r_2, s_2)$. To analyze how the bids in $e_1$ affect its allocation outcome, we fix the bid for $e_2$ at (0,0), (0.25,0.25), (0.5,0.5), and (0.75,0.75) and observe the impact of the bids from $r_1$ and $s_1$ in $e_1$.

As shown in Figure 3, we present the allocation results of BundleNet and JRegNet in two experimental settings. We observe that BundleNet exhibits clearer boundary awareness than JRegNet, accurately identifying scenarios where Bundle $e_1$ should not be allocated.

To better demonstrate BundleNet's capability in learning the optimal mechanism, we compare its deterministic allocation

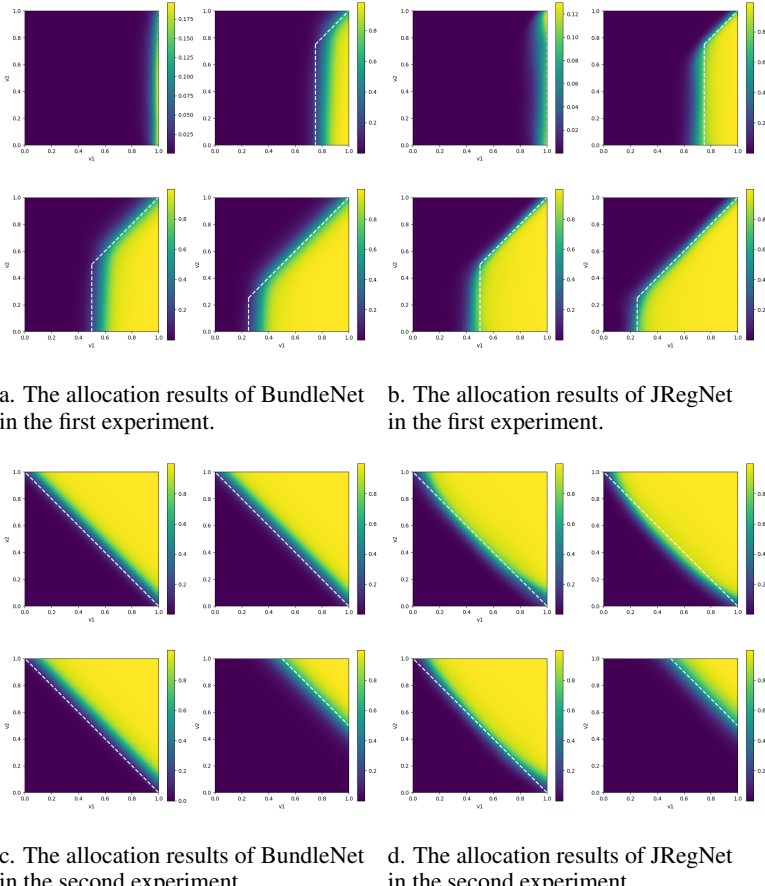

a. The allocation results of BundleNet in the first experiment.

b. The allocation results of JRegNet in the first experiment.

c. The allocation results of BundleNet in the second experiment.

d. The allocation results of JRegNet in the second experiment.

*Figure 3.* The figure presents the allocation rules learned by BundleNet and JRegNet under the **Setting** $U_2$. Subfigures (a) and (c) show the allocation results of BundleNet in two experiments, while (b) and (d) display the results of JRegNet in the same experiments. The solid regions in all subfigures depict the probability of the single-slot being allocated to bundle $e_1$, with the white dashed line representing the boundary of the allocation results derived from Myerson-like mechanism.

results with those RegretNet-like methods and the theoretical optimum in Figure 4. Unlike stochastic allocation, deterministic allocation provide clearer visual differentiation between these mechanisms. This visualization confirms BundleNet's superior approximation of the optimal mechanism, while JRegNet exhibits inconsistent performance. The deterministic view eliminates random noise, making the comparative advantages more apparent.

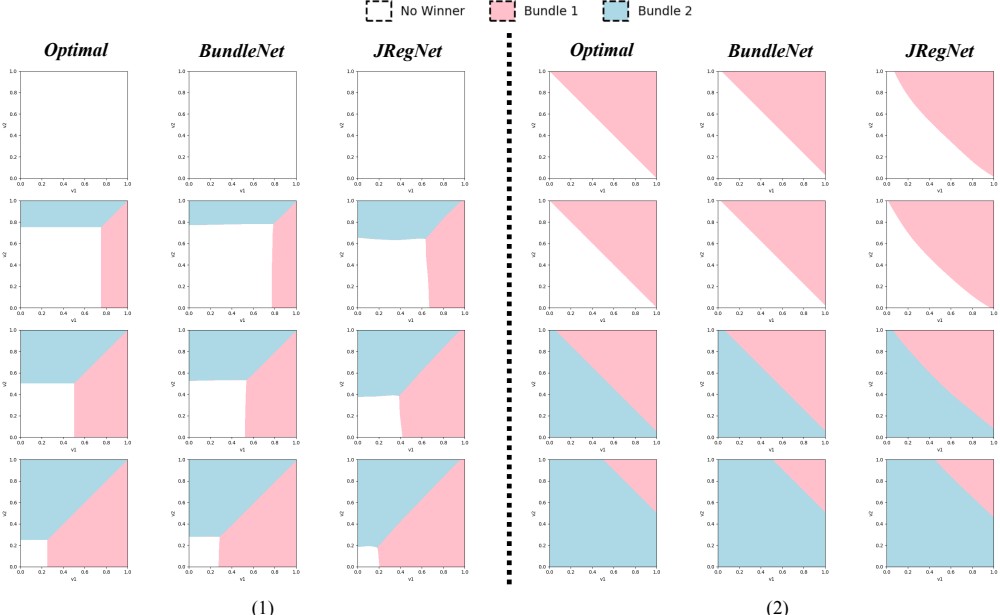

*Figure 4.* The figure presents the allocation rules learned by BundleNet and JRegNet under the **Setting** $U_2$. Subfigure (1) show the allocation results of optimal mechanism, BundleNet and JRegNet in the first experiment, while subfigure (2) display the results of three mechanism in the second experiment.

