# OpenReview forum: "Optimal Auction Design in the Joint Advertising"
_ICML.cc/2025/Conference — ICML 2025 poster_

### Official Review · Reviewer_X7Et · 2025-03-03

**Overall Recommendation:** 4

**Summary:**

This paper studies the auction design problem in the joint advertising scenario. For the single-slot setting, the optimal mechanism is derived. For the multi-slot setting, the authors propose BundleNet which achieves good performance.

**Claims And Evidence:**

I think so.

**Essential References Not Discussed:**

I am not really sure.

**Experimental Designs Or Analyses:**

It seems a bit strange to me that the authors bold all results of BundleNet in Table 1, even though they are not optimal in all cases.

**Methods And Evaluation Criteria:**

Generally yes. However, it seems to me that for the multi-slot case, the major improvement of this work is to model the IC constraint from a bundle's perspective rather than an agent's perspective. As for me, it seems a bit unclear why this works. However, it is sufficient for an applied paper. T

**Other Comments Or Suggestions:**

No.

**Other Strengths And Weaknesses:**

No.

**Questions For Authors:**

Please see the above.

**Relation To Broader Scientific Literature:**

The key contributions are related to automated mechanism design.

**Theoretical Claims:**

I didn't check the proof. Yet in Theorem 4.3, why do the authors write the theorem from the agent's perspective, rather than the bundle's perspective? The current writing makes understanding a bit hard, since the final allocation function is defined on the bundle, right? Meanwhile, $v_0$ appears in page 3, right, line 145, but not introduced before that. The same circumstance also appears for $U_i(v_i, v'_i)$.

---

> ### Author Rebuttal · Authors · 2025-04-01
>
> **Q1.** We agree this is an excellent suggestion. In fact, we noticed other reviewers raised similar questions. Here we clarify our key findings:
>
> 1. **Single-slot scenario**:
>          BundleNet achieves allocations **closer to the optimal solution** compared to baseline methods (as demonstrated in **Table 1**). Regarding the single-slot scenario, we have added a new visualization to comprehensively demonstrate BundleNet's capabilities. Based on the original experimental setup, we took two well-trained models (with average individual  regret values below 0.001, where the current JRegNet even shows higher revenue than BundleNet) and made their allocation results deterministic rather than randomized. The results show that BundleNet's allocations align more closely with the optimal solution.Here is the figure:https://anonymous.4open.science/r/BundleNet-6E3E/visualpicture.jpg
>
> 2. **Multi-slot scenario**:
>          BundleNet consistently yields **the highest revenue** among all compared approaches (shown in **Table 2**).
>
> For the camera-ready version, we will follow conventional practice by:
>
>    - **Bolding** the highest-revenue method in all tables
>    - Adding explicit textual markers (e.g., "★best") where appropriate
>    - Including a footnote to explain these annotations
>
>    This will eliminate potential ambiguity while maintaining rigorous comparison standards.
>
> **Q2.** I think this is a good question that is innovative enough for the article. Let me provide a more systematic response:
>
> 1. We propose **two novel solutions** for optimal mechanism design in joint advertising, and **empirically verify** the consistency between neural network results and theoretical outcomes.
>
> 2. For automated mechanism design, we present **a new approach** that differs from traditional neural network modifications. Our core methodology is as follows: By **reformulating the problem structure** and **optimization process**, we circumvent neural networks' limitations in handling sparse graph data, thereby improving performance.

---

### Official Review · Reviewer_EM98 · 2025-03-06

**Overall Recommendation:** 3

**Summary:**

The paper considers the sponsored search auctions with joint advertising. In this setting, advertisers are partitioned into suppliers and retailers, with a bipartite graph representing whether it is possible for the pair of advertisers to form a joint advertisement. Each advertisement is then assigned to a retailer and a supplier. The goal is to maximize the revenue of the platform, under incentive compatibility and individual rationality constraints. The authors first study the single-item case. Assuming monotone hazard rate value distributions of the advertisers, they propose an optimal mechanism for this case. The construction of the optimal mechanism is an adaptation of Myerson's result for single-item auctions without joint advertising. For the general case, the authors propose a neural network-based method to design the mechanism. The authors conduct experiments on synthetic data to show the effectiveness of the proposed method.

## Update after rebuttal

I stand by my original positive evaluation.

**Claims And Evidence:**

I think the claims made in the submission are supported by clear and convincing evidence.

**Essential References Not Discussed:**

I am not aware of any essential references that are missing.

**Experimental Designs Or Analyses:**

In general, the experimental design is solid and provides a good explanation for the effectiveness of the proposed method by comparing it with both previous method (JRegNet) and an fully IC mechanism. (RVCG). However, I have the following comments for the presentation of the experimental results: (1). Normally, when people make data points bold in tables, it means that the data point is the best among the methods. However, as far as I can see, in this paper, the bold data points indicate the method proposed in this paper. To see this, one could refer to Table 1. For $U_2, U_3, E_2, E_3$, the bold data points are not the best among the methods. This might give the reader a wrong impression. (2). The value distributions of the advertisers in the experiments of Sections 6.1 and 6.2 are different. I could not find the reason for this difference.

**Methods And Evaluation Criteria:**

The proposed methods and evaluation criteria make sense.

**Other Comments Or Suggestions:**

- Line 123 (right): The notation $E_i$ is used without definition.
- Line 153 (right): Why the expression of $\mathrm{SW}$ does not contain the term $v_0$?
- Line 364 (right): Missing line break before "Baseline".
- Line 403 (left): "::" should be ":".

**Other Strengths And Weaknesses:**

- This paper is overall solid and well-written.
- Like other papers in the automated mechanism design literature, the method's scalability beyond small instances is not good.

**Questions For Authors:**

- Do you have a reason for the difference in the value distributions of the advertisers in the experiments of Sections 6.1 and 6.2?

**Relation To Broader Scientific Literature:**

This paper is a nice contribution to the automated mechanism design literature.

**Theoretical Claims:**

The paper does not claim strong theoretical results. The optimality in the single-item case is an adaptation of Myerson's result, whose correctness is very intuitive, and the neural network-based method in the general case is a heuristic.

---

> ### Author Rebuttal · Authors · 2025-04-01
>
> **Q1.** To comprehensively validate our approach, we have conducted additional experiments across diverse settings:
>
> | Single-slot ｜LN(0.1,1.44) | $LN_2$     | $LN_3$     | $LN_4$     | $LN_5$     |
> | -------------------------- | ---------- | ---------- | ---------- | ---------- |
> | JRegNet                    | 0.9666     | 1.2519     | 1.2044     | 1.2415     |
> | BundleNet                  | 1.0132     | 1.2634     | **1.4042** | 1.4339     |
> | RVCG                       | 0.6337     | 0.9982     | 1.2388     | 1.4191     |
> | Optimal                    | **1.0227** | **1.2786** | 1.3916     | **1.4358** |
>
> | Single-slot ｜N(0.5,0.1) | $N_2$      | $N_3$      | $N_4$      | $N_5$      |
> | ------------------------ | ---------- | ---------- | ---------- | ---------- |
> | JRegNet                  | **0.8183** | **0.9091** | 0.8747     | 0.9117     |
> | BundleNet                | 0.7750     | 0.8692     | 0.9134     | 0.9561     |
> | RVCG                     | 0.5492     | 0.8114     | 0.8956     | 0.9444     |
> | Optimal                  | 0.7789     | 0.8656     | **0.9188** | **0.9582** |
>
> | Multi-Slots ｜N(0.5,0.1) | JRegNet    | BundleNet  | RVCG   |
> | ------------------------ | ---------- | ---------- | ------ |
> | $N_{5\times5}$           | **2.2071** | 2.1393     | 1.3110 |
> | $N_{6\times5}$           | 2.3537     | **2.3690** | 2.0132 |
> | $N_{7\times5}$           | 2.4814     | **2.4914** | 2.3565 |
> | $N_{8\times5}$           | 2.4740     | **2.6287** | 2.5343 |
> | $N_{9\times5}$           | 2.5185     | **2.7020** | 2.6395 |
> | $N_{10\times5}$          | 2.4711     | **2.7916** | 2.7228 |
> | $N_{11\times5}$          | 2.4082     | **2.8428** | 2.7840 |
> | $N_{12\times5}$          | 2.3272     | **2.8841** | 2.8383 |
>
> | Multi-Slots ｜E(2) | JRegNet | BundleNet  | RVCG   |
> | ------------------ | ------- | ---------- | ------ |
> | $E_{5\times5}$     | 1.1590  | **1.1884** | 0.6379 |
> | $E_{6\times5}$     | 1.2435  | **1.3718** | 0.9726 |
> | $E_{7\times5}$     | 1.2054  | **1.5497** | 1.2436 |
> | $E_{8\times5}$     | 1.2620  | **1.7162** | 1.4775 |
> | $E_{9\times5}$     | 1.3269  | **1.7937** | 1.6675 |
> | $E_{10\times5}$    | 1.4370  | **1.9463** | 1.8347 |
> | $E_{11\times5}$    | 1.5389  | **2.1050** | 2.0016 |
> | $E_{12\times5}$    | 1.6002  | **2.1869** | 2.1189 |
>
> **Q2.** We agree this is an excellent suggestion. In fact, we noticed other reviewers raised similar questions. Here we clarify our key findings:
>
> 1. **Single-slot scenario**:
>          BundleNet achieves allocations **closer to the optimal solution** compared to baseline methods (as demonstrated in **Fig. 4**).
>
> 2. **Multi-slot scenario**:
>          BundleNet consistently yields **the highest revenue** among all compared approaches (shown in **Table 2**).
>
> For the camera-ready version, we will follow conventional practice by:
>
>    - **Bolding** the highest-revenue method in all tables
>    - Adding explicit textual markers (e.g., "★best") where appropriate
>    - Including a footnote to explain these annotations
>
>    This will eliminate potential ambiguity while maintaining rigorous comparison standards.
>
> **Q3.** We sincerely appreciate the reviewer's careful reading regarding the writing details. We agree that the social welfare (SW) formulation could be more precise. In the revised version, we will correct it to:
> $$
>  \text{SW} =E_{\mathbf{v} \sim F}\left[v_0\left(\mathbf{1}  -\sum_{e \in E}x^e(\mathbf{v})\right) \boldsymbol{\lambda}^T + \sum_{e=(r,s) \in E}x^e(\mathbf{v}) (v_r + v_s)\boldsymbol{\lambda}^T \right]
> $$

---

### Official Review · Reviewer_DGzw · 2025-03-11

**Overall Recommendation:** 3

**Summary:**

The paper addresses auction design in joint advertising settings, where both retailers and suppliers jointly bid on ad slots. It extends classical auction theory by identifying an optimal mechanism for single-slot joint advertisements (via a Myerson-inspired framework) and introduces BundleNet—a novel neural network architecture that leverages a bundle-centric, graph-based approach—for multi-slot scenarios. This method also incorporates a new bundle-level incentive compatibility (IC) constraint, enabling the model to capture the interdependencies between the two parties. Experimental results show that BundleNet not only approximates the theoretical optimum in single-slot settings but also outperforms existing methods in multi-slot environments while maintaining approximate IC and individual rationality.

**Claims And Evidence:**

The major claims of the paper are well-supported by theoretical analysis and experimental results:
- The claim of identifying an optimal mechanism for single-slot joint advertising is supported by a theoretical derivation extending Myerson's framework to the joint advertising scenario (Theorem 4.3).
- The claim that BundleNet approximates the theoretical optimal mechanism in single-slot settings is shown numerically in Table 1 and Figure 3.
- The claim of superior performance in multi-slot settings is supported by comparative experiments against RVCG and JRegNet across different distribution settings (Tables 2 and 3).
- The theoretical foundation for the bundle-centric approach relies on Lemma 5.1, which shows that bundle IC constraints provide an upper bound on individual IC constraints.

Questions and potential issues:
- For the last point, it remains unclear to me when reading Lemma 5.1 whether this approach might restrict the space of discoverable mechanisms, can the authors elaborate on this?
- The proposed framework convincingly outperforms the major baseline of JRegNet, especially in larger settings. However, Figure 3, where the authors visualize the allocation rules learned by BundleNet and JRegNet in a particular setting, show the two methods learning comparable allocation rules with only minor differences. This makes me wonder whether the improved performance could be due to optimization or hyperparameter issues. Can the authors provide any examples where learned allocation structure exhibits a noticeable difference over the baselines, or further justify the superiority of their method, ruling out optimization or hyperparameter tuning as confounding factors?

**Essential References Not Discussed:**

None major.

**Experimental Designs Or Analyses:**

The experimental design is robust, with comparisons against strong baselines such as VCG and JRegNet. However, as mentioned above, while the synthetic datasets serve well for controlled comparisons, for the specific problem of ad auctions, the realism of these experiments could be enhanced by incorporating more complex simulations or real-world data to better capture practical ad auction scenarios.

**Methods And Evaluation Criteria:**

The evaluation methods align well with current standards in automated mechanism design; however, the synthetic nature of the dataset may oversimplify the dynamics of real advertising environments, particularly in capturing complex bidder dependencies and realistic click-through rate behaviors. I don't view it as a major problem, but I feel this is worth mentioning. The work would benefit from validating the approach on more complex or real-world datasets to assess its robustness and practical applicability.

**Other Comments Or Suggestions:**

The symbols are dense at times, readability could be improved.

**Other Strengths And Weaknesses:**

See above sections.

**Questions For Authors:**

Some questions above are restated here for readability:
- Is the focus on bundle IC constraints without loss of generality, or might it restrict the space of discoverable mechanisms? Does the upper bound relationship in Lemma 5.1 extend to equivalence under certain conditions?
- Although BundleNet shows significant numerical performance improvements, the visual differences in allocation boundaries depicted in Figure 3 appear subtle. Could you provide specific examples or additional evidence demonstrating that structural characteristics of BundleNet's learned allocation rules (e.g., sharper or more distinct boundaries) directly contribute to its performance gains? Alternatively, if these structural differences are minimal, can you elaborate on how you rule out potential confounding factors—such as optimization issues or hyperparameter effects—as the primary drivers behind the observed improvements?
- How transferable is the bundle-centric approach and graph-based architecture to other mechanism design problems beyond joint advertising?
- Like RegretNet, BundleNet only approximates IC rather than guaranteeing it. Have you explored or have ideas for mechanisms to provide stronger theoretical guarantees on the degree of IC violation?

**Relation To Broader Scientific Literature:**

The paper effectively builds upon and advances several research streams, though with some limitations in transferability:
- It builds upon automated mechanism design approaches (Duetting et al., 2019) via neural network
- It advances work on joint advertising mechanisms (Ma et al., 2024; Zhang et al., 2024), specifically addressing limitations in existing approaches like JAMA and JRegNet.
- The approach shares RegretNet's limitation of only approximating IC rather than guaranteeing it, which remains an open problem in neural mechanism design.
- The transferability of the bundle-centric approach and graph-based architecture to other mechanism design problems remains unclear, as the approach is heavily tailored to the specific bipartite structure of joint advertising.

**Theoretical Claims:**

- Theorem 4.3 (optimal mechanism for single-slot joint advertisement) extends Myerson's framework to the joint advertising setting.
- Lemma 5.1 (relationship between bundle IC constraints and individual IC constraints) demonstrates that the sum of bundle IC constraints upper-bounds the sum of individual IC constraints.

I read through, but didn't check correctness in detail for the two proofs.

---

> ### Author Rebuttal · Authors · 2025-04-01
>
> **Q1.** This is a very good question. I think this method may limit the space of discoverable mechanisms, but the difference is not significant.
>
> 1. Our core idea is to improve performance by limiting the space of discoverable mechanisms, avoiding the neural network's processing of sparse graph data. In the RegretFormer paper, their improvement likely relies more on expanding the discoverable mechanism space to find a better mechanism. However, for the more challenging problem of joint auctions, our experiments show that as the number of bundles increases, the performance of JRegNet gradually worsens. The main reason for this could be the sparsity of the graph. As the number of bundles increases, the sparsity of the graph significantly strengthens. Encoding this into the neural network inevitably degrades performance.
>  2. We argue that the inequality in **Lemma 5.1** becomes an equality when the following condition holds: For every agent $i \in R \cup S$, the utility functions $u_i^e$ across all edges $e$ can be simultaneously maximized by the same strategy $v'_i$.
> 3. Our method does not significantly limit the space of discoverable mechanisms. In our training process, we observed that increasing the number of misreport points to 10 does not lead to a significant difference in the sum of individual regrets versus the sum of bundle regrets. Additionally, the result of BundleNet converges to approximately the theoretical optimal.
> 4. The inequality in **Lemma 5.1** is difficult to extend into an equality.
>
> **Q2.** Regarding the single-slot scenario, we have added a new visualization to comprehensively demonstrate BundleNet's capabilities. Based on the original experimental setup, we took two well-trained models (with individual average regret values below 0.001, where the current JRegNet even shows higher revenue than BundleNet) and made their allocation results deterministic rather than randomized. The results show that BundleNet's allocations align more closely with the optimal solution. Here is the figure:https://anonymous.4open.science/r/BundleNet-6E3E/visualpicture.jpg
>
> **Q3.** We conducted experiments using real-world data, including 10 bundles and 5 slots. The dataset consists of 102,679 samples, with 80% used for training and 20% for validation. JRegNet and BundleNet are both used in the fine sorting phase. During preprocessing, we normalized the bidding data to facilitate neural network learning. After obtaining the final results, we scaled the values back to the original bid price range. We evaluated the revenue of the three methods, and for methods with IC Violation, $\widehat{rgt}$  during testing on normalized data was less than 0.0005. The results are presented below:
>
> |      | RVCG    | BundleNet | JRegNet |
> | ---- | ------- | --------- | ------- |
> | Rev  | 26.7173 | 38.1939   | 36.2950 |
>
>    Note: In the real world data bidding data, there are actually fewer than 50% of the cases that involve more than 5 bundles. Instead of treating them separately, we use a single unified neural network to process all bundle sizes, which contributes to JRegNet’s superior performance over VCG.
>
> **Q4.** BundleNet's approach is very transferable:
>
> 1. While the idea of joint auctions originated in online advertising, it can be applied to more general scenarios, such as government procurement projects where a single project requires collaboration between companies from different domains. Extending this to allow more participants to form coalitions for joint bidding presents an interesting theoretical problem. Our modeling approach could be similarly adapted for such cases.
> 2. As demonstrated in an AAAI 2025 paper (building upon the new scenario introduced in KDD 2024's "Ad vs Organic: Revisiting Incentive Compatible Mechanism Design in E-commerce Platforms"), JRegNet has been successfully applied to merged ranking mechanism design, achieving excellent results.
> 3. We particularly emphasize our core research insight: When adapting RegretNet to broader mechanism design problems where neural networks struggle with certain features, our method of constraining the discoverable mechanism space can effectively improve performance.
>
> **Q5.** I consider this an open problem in automated mechanism design. Existing methods for multi-agent, multi-item auctions face various limitations:
>
>    - RegretNet can only guarantee approximate IC
>    - GemNet may suffer from over-allocation issues
>    - AMenuNet might not sufficiently approximate theoretical solutions
>
> From my perspective, the key may lie in characterizing feasible auctions through utility function properties (analogous to Rochet’s theorem for single-item cases). Similar to RochetNet’s approach, we could:
>
>    - Formalize such properties for multi-bidder settings
>    - Integrate them into neural architecture design
>    - Develop a new automated mechanism design framework
>
> We are currently investigating concrete implementations, though technical details remain under study.

---

### Official Review · Reviewer_BtzC · 2025-03-15

**Overall Recommendation:** 3

**Summary:**

This paper proposes an optimal mechanism for joint advertising in a single-slot setting and introduces BundleNet, a bundle-based neural network for multi-slot joint advertising. Through extensive experimentation, the paper demonstrates that BundleNet effectively approximates theoretical results in single-slot settings and outperforms existing methods in multi-slot scenarios.

**Claims And Evidence:**

yes

**Essential References Not Discussed:**

The essential references are discussed in this paper.

**Experimental Designs Or Analyses:**

The paper lacks experiments on real-world datasets.

**Methods And Evaluation Criteria:**

The optimal mechanism should be helpful to the bidding advertisement area.

**Other Comments Or Suggestions:**

1.  The abbreviation "IC" is introduced before the full term "incentive compatibility" is defined. It would be clearer to first provide the full term before using the abbreviation.
2. It would be helpful to provide more explanation about joint advertisement in the Figure 1 label, as the relevant background has not been clearly introduced.
3. the paper presents interesting findings, but the contribution is not sufficiently emphasized. It would be helpful to clarify how this work advances the field.

**Other Strengths And Weaknesses:**

Strengths：
1. The paper is well-written and is easy to follow.
2. The claims of this paper is validated via experiments
3. A detailed theoretical proof is provided, which adds rigor to the proposed approach and validates the claims made in the paper.

Weaknesses:
1. The study is validated only on a simulated dataset, which raises concerns about its practical applicability. Evaluating the method on real-world data would strengthen its reliability and impact.
2. The experimental settings are not clearly described, particularly regarding model parameters and related configurations. Providing more details would improve the reproducibility and clarity of the study.
3. The experiments appear relatively simple and may not be comprehensive enough. Including additional tests, such as parameter sensitivity analysis, would provide a more thorough evaluation of the proposed method.

**Questions For Authors:**

1. The study is validated only on a simulated dataset, which raises concerns about its practical applicability. Evaluating the method on real-world data would strengthen its reliability and impact.
2. The experimental settings are not clearly described, particularly regarding model parameters and related configurations. Providing more details would improve the reproducibility and clarity of the study.
3. The experiments appear relatively simple and may not be comprehensive enough. Including additional tests, such as parameter sensitivity analysis, would provide a more thorough evaluation of the proposed method.
4.  It would be helpful to provide more explanation about joint advertisement in the Figure 1 label, as the relevant background has not been clearly introduced.
5. the paper presents interesting findings, but the contribution is not sufficiently emphasized. It would be helpful to clarify how this work advances the field.

**Relation To Broader Scientific Literature:**

NA

**Theoretical Claims:**

Yes

---

> ### Author Rebuttal · Authors · 2025-04-01
>
> We would like to thank you for the constructive comments and suggestions! Below are our responses to your questions.
>
> **Q1.** We conducted experiments using real-world data from a leading Internet company. In practice, advertisements will go through the stages of recall, rough sorting, and fine sorting. A maximum of 10 advertisements and 5 advertising spaces will be reserved after rough sorting and will go into the fine sorting stage. JregNet and BundleNet are both used in the fine sorting phase. The dataset consists of 102,679 samples, with 80% used for training and 20% for validation. We compared the revenue differences among RVCG, JRegNet, and BundleNet. During preprocessing, we normalized the bidding data to facilitate neural network learning. After obtaining the final results, we scaled the values back to the original bid price range. We evaluated the revenue of these three methods, and for methods with IC Violation, $\widehat{rgt}$ during testing on normalized data was less than 0.0005. The results are presented below：
>
> |         | RVCG    | BundleNet | JRegNet |
> | ------- | ------- | --------- | ------- |
> | Revenue | 26.7173 | 38.1939   | 36.2950 |
>
> **Note:** In the real world data bidding data, in the fine sorting stage, there are actually less than 50% of the cases that involve more than 5 bundles. Instead of treating them separately, we use a single unified neural network to process all bundle sizes, which contributes to JRegNet’s superior performance over VCG.
>
> **Q2.** In our experiments, the hyperparameter settings are consistent with the open-source implementations of JRegNet and RegretNet.
>
> For both JRegNet and BundleNet, we use a neural network with 6 hidden layers, each containing 100 neurons, for training and testing. We initialize our Lagrange parameters as $\forall e \in E, \lambda_e = 2.0$. In each gradient ascent step, the update parameter for the Lagrange multiplier and the quadratic penalty term is set to $\rho = 2.0$. The learning rate for searching the optimal misreport value is $\gamma = 0.1$, and the neural network learning rate is $\eta = 0.001$. We use the Adam optimizer for all updates. The primary modification we made was adjusting the batch size for training both networks to 1024. We will release our code upon the acceptance of this paper to facilitate reproduction and further research.
>
> **Q3.** Based on the aforementioned hyperparameters, we present a sensitivity analysis of the batch size and the initial value of the Lagrange multiplier in the $U_2$ scenario：
>
>    - Sensitivity Analysis of Batch Size：
>
>      |           | 128    | 256    | 512    | 1024   | 2048   | 4096   |
>      | --------- | ------ | ------ | ------ | ------ | ------ | ------ |
>      | BundleNet | 0.4993 | 0.4971 | 0.5174 | 0.5262 | 0.5286 | 0.5295 |
>      | JRegNet   | 0.5535 | 0.5594 | 0.5641 | 0.5638 | 0.5685 | 0.5658 |
>
>    - Sensitivity Analysis of Initial Lagrange Multiplier
>
>      |           | 1.0    | 2.0    | 3.0    | 4.0    | 5.0    | 6.0    |
>      | --------- | ------ | ------ | ------ | ------ | ------ | ------ |
>      | BundleNet | 0.1480 | 0.5330 | 0.5373 | 0.5267 | 0.5229 | 0.5240 |
>      | JRegNet   | 0.5669 | 0.5659 | 0.5653 | 0.3837 | 0.1386 | 0.1165 |
>
> **Q4.** Thank you for your feedback. We will provide a more detailed description of the joint advertisement in the label for Figure 1.  The image on the left illustrates traditional advertising, where the **retailer** submits a bid for the ad, while the **supplier** (brand owner) does not participate in the auction. In this case, the platform only receives the bid from the **retailer**.  In contrast, as shown in the right image, for a joint advertisement, both the **retailer** and the **supplier** participate in the auction. They submit bids simultaneously, and the platform receives bids from both parties for the same ad.
>
> Thank you for recognizing our work. Here we re-emphasize the key contributions and findings of our work:
>
> 1. **Novel solution**: We propose a novel solution for automated mechanism design in joint advertising markets, and empirically demonstrate the alignment between neural network outputs and theoretical predictions.
> 2. **Better performance**: For automated mechanism design, we introduce a fundamentally different approach from conventional neural network adaptations. Our core methodology works by **reformulating the problem structure** and **redesigning the optimization process**, thereby overcoming neural networks' inherent limitations in processing sparse graph data while achieving performance improvements.
> 3. **Interesting Findings** : We theoretically and empirically show that directly applying RegretNet-like methods to new problem domains may fail to discover optimal mechanisms, as significant gaps can persist between learned mechanisms and theoretically optimal solutions. Our analysis demonstrates that successful application requires problem-specific modifications with rigorous theoretical validation.

---

> > ### Comment · Reviewer_BtzC · 2025-04-05
> >
> > I confirm that I have read the author response to my review and will keep my recommendation score.

---

> > > ### Author Response · Authors · 2025-04-08
> > >
> > > Dear Reviewer BtzC,
> > >
> > > We extend our sincerest gratitude for the invaluable time and meticulous effort you dedicated to reviewing our paper. We keenly want to know whether our responses have effectively resolved your initial concerns. If there are any remaining questions or if further clarification is required, we would be delighted to engage in further discussion to ensure that all your concerns are thoroughly addressed.
> > >
> > > Again, thank you for your dedication to scientific rigor and for helping us refine our paper.
> > >
> > > Best regards，
> > >
> > > The Authors

---

### Decision · Program_Chairs · 2025-05-01

**Decision:**

Accept (poster)

**Comment:**

The joint advertising problem arises in sponsored search auctions where two parties, typically a supplier and a retailer, jointly benefit from an ad impression and co-bid for the same ad slot. The paper proposes an optimal mechanism for single-slot joint advertising, and introduces BundleNet, a graph-based neural architecture for multi-slot settings that incorporates bundle-level IC constraints. Theoretical contributions are well-motivated and extend classical auction design. BundleNet outperforms baselines on synthetic data and approximates theoretical optima.

On the negative side, all experiments are on simulated data, limiting practical applicability. The bundle IC constraint’s impact is not fully clarified, and experimental details are sometimes unclear. Scalability and generalizability remain open questions. Despite these issues, the paper is well-written, and is a solid contribution to the sponsored search auction literature, so it can be accepted if there is room.